# Evaluation of systems reform in public hospitals, Victoria, Australia, to improve access to antenatal care for women of refugee background: An interrupted time series design

Jane Yelland[1,2]*, Fiona Mensah[3,4], Elisha Riggs[1,2], Ellie McDonald[1], Josef Szwarc[5], Wendy Dawson[1], Dannielle Vanpraag[1], Sue Casey[5], Christine East[6,7], Mary Anne Biro[6], Glyn Teale[8], Sue Willey[6], Stephanie J. Brown[1,2,4]

1 Intergenerational Health, Murdoch Children's Research Institute, Parkville, Victoria, Australia, 2 Department of General Practice, The University of Melbourne, Melbourne, Victoria, Australia, 3 Clinical Epidemiology and Biostatistics Unit, Murdoch Children's Research Institute, Parkville, Victoria, Australia, 4 Department of Paediatrics, The University of Melbourne, Melbourne, Victoria, Australia, 5 Victorian Foundation for Survivors of Torture, Brunswick, Victoria, Australia, 6 School Nursing and Midwifery, Monash University, Clayton, Victoria, Australia, 7 School of Nursing and Midwifery, Mercy Health and La Trobe University, Bundoora, Victoria, Australia, 8 Women's and Children's, Western Health, Sunshine, Victoria, Australia

* jane.yelland@mcri.edu.au

**Data Availability Statement:** The routinely collected perinatal data analysed in the current

## Abstract

### Introduction

Inequalities in maternal and newborn health persist in many high-income countries, including for women of refugee background. The Bridging the Gap partnership programme in Victoria, Australia, was designed to find new ways to improve the responsiveness of universal maternity and early child health services for women and families of refugee background with the codesign and implementation of iterative quality improvement and demonstration initiatives. One goal of this 'whole-of-system' approach was to improve access to antenatal care. The objective of this paper is to report refugee women's access to hospital-based antenatal care over the period of health system reforms.

### Methods and findings

The study was designed using an interrupted time series analysis using routinely collected data from two hospital networks (four maternity hospitals) at 6-month intervals during reform activity (January 2014 to December 2016). The sample included women of refugee background and a comparison group of Australian-born women giving birth over the 3 years. We describe the proportions of women of refugee background (1) attending seven or more antenatal visits and (2) attending their first hospital visit at less than 16 weeks' gestation compared over time and to Australian-born women using logistic regression analyses.

study has been retrieved from the participating hospitals under the approval of the Human Research Ethics Committees (HREC) of the Royal Children's Hospital (Melbourne, Australia), Monash Health and Western Health. Requests for data access should be made to the Human Research Ethics Committee of the Royal Children's Hospital, rch.ethics@rch.org.au, in the first instance.

**Funding:** This work was supported by the National Health and Medical Research Council of Australia https://www.nhmrc.gov.au/ (ID 1056799). JY was supported by a National Health and Medical Research Council Career Development Fellowship (ID 1062484, 2013-2017) and a Translating Research into Practice Fellowship (ID 1150566, 2018-2019). SB was supported by an Australian Research Council https://www.arc.gov.au Future Fellowship (ID 110101036, 2012–2015) and a National Health and Medical Research Council Research Fellowship (ID 1103976, 2016–2020). The funders had no role in study design, data collection and analysis, the decision to publish, or preparation of the manuscript.

**Competing interests:** The authors have declared that no competing interests exist.

**Abbreviations:** adjOR, adjusted odds ratio; BOS, Birthing Outcome System; CI, confidence interval; GP, general practitioner; OECD, Organisation for Economic Co-operation and Development; STROBE, Strengthening the Reporting of Observational Studies in Epidemiology..

In total, 10% of births at participating hospitals were to women of refugee background. Refugee women were born in over 35 countries, and at one participating hospital, 40% required an interpreter. Compared with Australian-born women, women of refugee background were of similar age at the time of birth and were more likely to be having their second or subsequent baby and have four or more children. At baseline, 60% of refugee-background women and Australian-born women attended seven or more antenatal visits. Similar trends of improvement over the 6-month time intervals were observed for both populations, increasing to 80% of women at one hospital network having seven or more visits at the final data collection period and 73% at the other network. In contrast, there was a steady decrease in the proportion of women having their first hospital visit at less than 16 weeks' gestation, which was most marked for women of refugee background. Using an interrupted time series of observational data over the period of improvement is limited compared with using a randomisation design, which was not feasible in this setting.

## Conclusions

Accurate ascertainment of 'harder-to-reach' populations and ongoing monitoring of quality improvement initiatives are essential to understand the impact of system reforms. Our findings suggest that improvement in total antenatal visits may have been at the expense of recommended access to public hospital antenatal care within 16 weeks of gestation.

## Author summary

### Why was the study done?

- Women of refugee background have rates of stillbirth and perinatal mortality two to three times higher than Australian-born women.

- Antenatal care is a key preventive strategy for the optimal health of pregnant women and newborn babies and is critical to addressing modifiable factors for poor health outcomes.

- Refugee women experience barriers in accessing and engaging in antenatal care, and healthcare providers report challenges in responding to the social context and needs of refugee families.

### What did the researchers do and find?

- We codesigned and implemented multiple quality improvement and demonstration initiatives in universal health services, including four maternity hospitals.

- We used an interrupted time series design to assess the timing and number of antenatal clinical visits for refugee women compared with Australian-born women over 3 years of reform.

- Applying a method devised by the partnership, we identified from routinely collected data that 10% of all women giving birth at the participating hospitals were of refugee background.

- There was an increase over time in the proportion of refugee and Australian-born women attending the standard number of visits, with women of refugee background commencing hospital antenatal care well past the recommended gestation.

## What do these findings mean?

- For women of refugee background, delayed access to hospital-based care increases the likelihood of missing out on critical elements of pregnancy care vital to optimising maternal and child outcomes.

- Accurate ascertainment of this 'harder-to-reach' population is essential to evaluating the impact of quality improvement initiatives on refugee families' access to healthcare.

- An interrupted time series design with a comparison group is suited to measuring change in number and timing of antenatal visits, although data quality, contextual influences, and qualitative insights into system change are important for interpreting outcomes.

- Ongoing monitoring of quality improvement initiatives on populations vulnerable to poor health outcomes is essential to understand the impact of system reforms and efforts to reduce inequalities.

## Introduction

Antenatal care is universally accepted as a key preventive strategy for the optimal health of pregnant women and newborn babies [1,2]. Clinical guidelines in the United States, United Kingdom, and Australia recommend commencement of antenatal care in the first trimester of pregnancy and a minimum of seven pregnancy visits, involving assessment and promotion of maternal physical and psychological health and well-being, in addition to screening and intervention for serious medical conditions and pregnancy complications [3,4].

Although Australia compares favourably to other Organisation for Economic Co-operation and Development (OECD) countries with regard to maternal and newborn health outcomes [5], women of refugee background have rates of stillbirth and perinatal mortality two to three times higher than Australian-born women [6–8]. Several studies shed light on contributing factors, including communication and language barriers, impacts of trauma, flight from war and persecution in countries of origin, and challenges of settlement in a new country [9–11]. Qualitative research with refugee families living in Melbourne, Australia, has identified the significant challenges families experience when navigating Australian maternity and primary care services, including not knowing what to expect, having limited understanding of pregnancy services or routine antenatal tests and procedures, difficulties with transport to get to appointments, and limited access to professional interpreters [12]. A parallel study investigating the experiences of obstetricians, midwives, and other health professionals working in

public maternity services in the same region found that they experience a range of challenges responding to the complex needs and social circumstances of refugee families and frequently lack confidence in how to work with families experiencing trauma or with low health literacy [12].

Evidence from this study was a catalyst for the establishment of the Bridging the Gap partnership, which brought together clinicians, managers, policy makers, and researchers to find new ways to improve the responsiveness of universal maternity and early child health services for women and families of refugee background. The partnership worked together over 4 years to support the codesign and implementation of iterative quality improvement and demonstration initiatives in maternity hospitals and maternal and child health clinics.

We hypothesised that these partnership-driven strategies would foster a more responsive service system and build service capability to promote better access to healthcare for women of refugee backgrounds and their families around the time of having a baby [13]. The objective of this paper is to investigate and report the numbers (and proportions) of women of refugee background at participating study hospitals who (1) attended their first hospital-based antenatal visit during the early second trimester (before 16 weeks' gestation) and (2) had at least seven hospital-based antenatal visits during pregnancy compared with Australian-born women at the same hospitals, during the establishment and implementation of the initiatives of the Bridging the Gap programme (2014–2016).

## Methods

The RECORD checklist for observational studies (as an extension of the STROBE guidelines) [14] was used to ensure all aspects of the research, when appropriate, had been reported (see S1 RECORD Checklist).

The study was conducted according to a prospective protocol including planned analysis of access to antenatal visits using the interrupted time series method [13].

### Context and setting

The setting for the programme initiatives is four publicly funded metropolitan maternity hospitals and two community-based early childhood health services in Victoria, Australia. The hospitals are managed by two health networks. Leaders from the networks 'self-selected' to participate in Bridging the Gap given the demographics of the population they served, concern about poor outcomes for women from diverse backgrounds, and commitment to improvement strategies in providing care to families of refugee background. One network comprises three hospitals (Monash Medical Centre, Dandenong Hospital, Casey Hospital), one of which is a tertiary referral centre with >8,000 births per annum (hereafter referred to as 'hospital network X'), and the other includes a single maternity hospital (Sunshine Hospital) with >5,000 births each year ('hospital network Y'). The networks are in different regions with diverse and rapidly growing refugee populations.

The universal health system in Australia covers healthcare costs in public hospitals, including maternity care. Women can select to have their pregnancy, birth, and postnatal care at any public hospital, although the reality is that most women opt to have care at the maternity hospital within the geographical area in which they live. Women are assigned (or select) a specific model of public maternity care. The majority of women booked to give birth at the participating hospitals attend outpatient antenatal clinics with a team of obstetric and midwifery staff. Women's entry into the public maternity system usually requires a referral from a general practitioner (GP, family physician). At the time of pregnancy confirmation, GPs will discuss with women options for preferred maternity hospital, taking into account clinical

considerations, e.g., existing medical conditions and previous pregnancy complications. It is increasingly common for hospitals to encourage women to attend GPs for early pregnancy care and for them to make arrangements for women to have early pregnancy screening tests.

The Bridging the Gap partnership adopted the term 'refugee background', acknowledging that people seeking asylum, refugees, and those who have come to Australia from humanitarian source countries but on other visa categories (e.g., family reunion) have been through similar experiences in their country of origin (or transit country) and will share common challenges in settlement, including accessing and engaging in healthcare.

## Bridging the gap initiatives

The aims of the Bridging the Gap programme were to implement and evaluate codesigned intervention strategies in maternity hospitals and early childhood health services to improve access to universal healthcare for families of refugee background, build organisational and system capacity to identify and address modifiable risk factors for poor maternal and child health outcomes in refugee populations, and develop a sustainable framework for ongoing quality improvement in responding to the needs of families of refugee backgrounds. Informed by Greenhalgh and colleagues' [15] model for innovation, Bridging the Gap took a 'whole-of-system' approach to improving care with multiple iterative improvement strategies implemented across the participating health services [13]. These are outlined in Table 1.

Strategies included new approaches to reforming data systems to improve ascertainment of women of refugee backgrounds [16,17]; a multiagency, community-informed model of group pregnancy care [19]; and initiatives to enhance the engagement of professional interpreters [18]. Partnership-designed and cofacilitated professional development activities took place over the period of implementation. Each initiative had objectives, target metrics, and milestones.

The multidisciplinary, multiagency partnership met in their regions frequently to consider local priorities for change and chart new ideas and next steps toward codesign. This included the formation of working groups for each improvement strategy. The research team (JY, ER, WD, DV) facilitated partnership meetings and working group sessions, took notes, and maintained feedback processes and partnership communication. Conversations between partners, protocol development, and data agreements were well underway in 2014, and most of the preparation, testing, refining, and evaluation of improvement strategies took place in 2015–2016.

## Study design

Interrupted time series using routinely collected data from maternity hospitals at 6-month intervals during reform activity (2014–2016) [13].

## Data collection procedures

Routinely collected maternal, perinatal, and service data were collected by hospitals and recorded in an electronic database called the Birthing Outcome System (BOS). This database is used to record information about all births at ≥20 weeks' gestation (including maternal details) at the participating hospitals. Hospital networks made data for selected items available to the research team in nonidentifiable format.

To contextualise routinely collected data, the research team also recorded changes to the organisation of maternity care that were external to the remit of Bridging the Gap, including the introduction of new policies and guidelines and training offered to hospital-based staff

**Table 1. Summary of multifaceted improvement initiatives.**

| Description of intervention | Health sector/agency | Rationale | Process/outcome |
|---|---|---|---|
| **Identifying women and families from refugee backgrounds** | | | |
| Adding 'year of arrival' to administrative data systems | Hospital X and hospital Y | Together with maternal country of birth, year of arrival provides a better proxy measure of likely refugee background. Ethnicity data are limited in identifying 'refugee background'; data on race less salient in Australian context and not used. | Data item added to maternity databases at participating hospital systems from 2015 [16] |
| Examine GP referral information indicating women of refugee background | Hospital X | Assess completion of four items to identify women of refugee background and service needs: country of birth, year of arrival, language spoken, interpreter required | Audit of referrals indicated good completion of items identifying refugee-background women; poorer completion of language items for other women; feedback to GPs, 2016 |
| Enhancing GP referrals of refugee women to maternity care | Hospital Y | Referral information from primary care important to enable appropriate triaging to appropriate model of care | Multicomponent improvement initiative with better completion of key data items, 2015–2016 [17] |
| Identifying refugee background clients in clinical encounter using questions developed by Bridging the Gap partners | MCH service | Improve ascertainment of refugee background as first step in providing care responsive to complex needs | Testing a set of questions with MCH nurses, 2015 [16] |
| Identifying refugee-background women for triage to caseload midwifery care | Hospital Y | Models offering continuity of care provider/s most likely to offer relational care responding to women's complex social circumstances | Attention to data issues and organisational restructure at the time of the initiative resulted in progress slowing, 2015–2016 |
| **Professional development** | | | |
| Professional development for clinicians, managers, front-of-house staff | Hospital X, hospital Y, and MCH service | Build workforce confidence and capacity in 'doing things differently' in supporting families of refugee background | Multiple sessions: the refugee and asylum seeker experience[#]; identifying women of refugee background; incidental counselling; working with interpreters; safety planning (family violence); case management for clients with socially complex needs, 2015–2017 |
| **GPC for women and families of refugee background** | | | |
| Multiagency GPC for refugee women provided by a multidisciplinary team | Hospital Y, MCH services; additional maternity hospital joined the GPC collaboration in 2016 | An innovative model of pregnancy and early postnatal care to improve women's access to care, reduce social isolation, and enhance health literacy | Implementation commenced in 2015, with qualitative evaluation demonstrating cultural safety and belonging for women of refugee background attending GPC [18] |
| **Engaging professional interpreters** | | | |
| Improving women's access to interpreter services in birth suites | Hospital X | Formative research indicated low use of professional interpreters in labour | Multifaceted initiative resulting in significant improvement of interpreter engagement for women in labour, 2015 [19] |
| System reform to enable access to language services at time of induction of labour | Hospital Y | Address clinician concern that women with low English proficiency were missing out on information about induction process | Scoping by working group and initial round of data collection from clinicians, 2015–2016. Progress stalled while service waited for new booking system for inductions |
| **Intersector collaboration supporting refugee families transitioning from maternity to MCH services** | | | |
| Maternity to MCH services programme to optimise the engagement of families transitioning from hospital to postpartum primary care | Hospital X and MCH service | Fragmentation of service sectors results in women of refugee background and their families 'falling through the gap' between hospital and postpartum maternal and child healthcare | Three projects testing referral pathways from maternity to early childhood health services and ways to optimise refugee women to continuity of care models, 2015–2016: (1) ascertainment of refugee-background pregnant women and triage to refugee antenatal clinic; (2) identification of refugee-background women in early postnatal care and referral to MCH refugee specialist team; and (3) identification of pregnant refugee-background women with additional children aged 0–6 not previously linked to MCH services and provided referral for older children |

[#]'Refugee and asylum seeker experiences' professional development sessions provided by specialist facilitators from partner agency, Victorian Foundation for Survivors of Torture.

Abbreviation: GP, general practitioner; GPC, group pregnancy care; MCH, maternal and child health

during the study period. The team also worked with members of the partnership and programme champions to identify contextual influences.

## Study population

The study population of primary interest comprised migrant women of refugee background giving birth at participating hospitals during the study period, with Australian-born women giving birth at the four hospitals during the same time period as the comparison group.

There is not a straightforward approach to identifying people of refugee background in Australian administrative data sets. There is no single 'refugee' visa in Australia, people may choose not to identify as a refugee once issued with protection visas, and there are sensitivities in asking women questions about migration background for administrative purposes. People may be reluctant to disclose their migration history for fear of how this information may be used. Hence, it is problematic for services to determine 'visa status,' and use of this information is likely to result in underascertainment [16,20].

Country of birth was considered the best available proxy measure for identifying women of refugee background in routinely collected hospital data systems. Information on maternal country of birth was combined with information collected by the Australian Department of Social Services [21], which enabled us to identify countries from which more than two-thirds of entrants to Australia came via a humanitarian entry pathway over the 10 years prior to the study period (2014–2016). An exception to this is when country of birth is combined with knowledge of a person's preferred language. Using these two items together can identify language-specific minority ethnic/cultural groups that are likely to be of refugee background.

Initially, data on year of arrival in Australia were not available in routinely collected hospital data. The addition of a new data item, 'year of arrival in Australia', was gradually included in the data systems from 2015 as a Bridging the Gap initiative [16,17]. As this information became available, we used this to describe the migration characteristics of the sample and consider number of antenatal visits by women's recent arrival in Australia.

## Migration and obstetric characteristics

Migration characteristics, including country of birth (name of country), language spoken (name of language), and interpreter required (yes/no), were identified for women of refugee background. Obstetric characteristics, including maternal age (mean, standard deviation; categorical 14–19, 20–24, 25–29, 30–34, 35+ years), parity (expecting first to fourth or later baby), gravida (women having their first pregnancy up to their fourth or later pregnancy), gestational diabetes (yes/no), and hypertension/preeclampsia (yes/no), were similarly identified for women of refugee background and Australian-born women.

## Outcomes

The primary outcome of interest prespecified in the study protocol [13] is number of hospital-based antenatal visits, specifically women having seven or more visits as per Australian guidelines.

A secondary outcome is gestation of pregnancy at first hospital-based antenatal visit. Evidence-based guidelines recommend that the first antenatal visit take place in the first trimester of pregnancy, considered to be up to 14 weeks' gestation. We considered gestation of up to 16 weeks as indication that timing beyond this point falls well short of the guidelines.

Both study sites record hospital antenatal visits and gestation at each visit in BOS. Visits are defined as visits to a hospital practitioner/pregnancy clinic. Visits for serum screening, diagnostic testing, or imaging are not classified as antenatal visits.

## Target sample size

Our initial power calculations were based on a sample size of 350 women in each 6-month period. Based on this sample size, we estimated the study would have 88% power with alpha of 0.05 to detect a halving of the proportion of women attending less than seven visits from 30% to 15% accounting for clustering of women within hospitals (intraclass correlation in outcome of 0.05). At the time of developing the study protocol (2013), a total of 14,000 women per annum were giving birth at the four participating hospitals, and we conservatively estimated that around 700 (5%) of these women were of refugee background [13].

## Analyses

Analysis presented differed to that of the protocol [13], with additional data not previously expected (enabling a more robust adjustment for clustering) and timing of access to antenatal care (from the first antenatal visit at 14 weeks' gestation in the protocol to 16 weeks in the analysis) as a measure of care commencing well past the recommended first trimester of pregnancy.

Data were analysed using Stata version 15 [22]. Migration characteristics were described for women of refugee background and by hospital network, and their obstetric characteristics were compared with Australian-born women. The mean (and range) and number of visits were examined for women of refugee background and Australian-born women according to hospital network and the number of weeks pregnant (gestation) when the baby was born.

In line with the interrupted time series design, the proportion of women of refugee background who attended seven or more antenatal visits and women who attended their first antenatal visit at less than 16 weeks' gestation were compared at 6-month intervals over the 3 years. Comparisons of the number of visits (seven or more) and gestation at first visit (<16 weeks) were made with Australian-born women over the course of the introduction of the intervention strategies. Logistic regression analyses were conducted within each setting, hospital network X and hospital network Y, accounting for the clustering effects of women's country of birth within each hospital network using robust estimation of standard errors. Analytic models included an indicator of whether the time period was pre- or postimplementation and a linear trend for time to account for potential intervention effects while taking account of any 'secular' trends in outcomes that may have been occurring over the baseline and intervention period. Covariates of interest included parity and gestational diabetes. Interactions were tested between refugee background and each of the time period and trend effects, parity, and gestational diabetes. Regression analyses were replicated using 3-month intervals over the 3 years of observation to meet the criteria of at least 8 data points recommended by Penfold and Zhang [23] (increasing the number of data points to 11 for hospital network X and 9 for hospital network Y). Minimal differences to the study findings resulted, and analysis using 6-month intervals is presented to clarify comparisons between women of refugee background and Australian women at each time point.

## Ethics statement

Human research ethics approval was obtained by the Human Research Ethics Committee of the Royal Children's Hospital (Approval 33179), Monash Health (Approval 14318X), Western Health (Approval 33179A), and Victorian Department Education and Training (Approval 2014_002513). Individual consent was not obtained because routinely collected hospital data were released to the research team deidentified and analysed anonymously.

## Results

### Characteristics of the sample

Over the 3 years of the Bridging the Gap programme, the number of women giving birth at the four hospitals increased from around 14,000 in 2014 to 15,000 in 2016. Our decision-making to derive the refugee background sample taking maternal country of birth and completeness of other migration characteristics is illustrated in a flowchart in S1 Fig.

Table 2 presents the migration characteristics of the sample by hospital network.

In total, 60% of the women giving birth at the four hospitals comprising the two hospital networks were migrants, the majority coming from countries where English is not the main language. Over 10% of the sample was identified as being of refugee background, significantly more than projected in our initial estimates.

Women of refugee background at the two hospital networks (Table 2) were born in over 35 countries. Just over half of the women of refugee background attending hospital network X were born in Afghanistan, and more than one-third of the women of refugee background attending hospital network Y were born in Sudan. Given the benefit of having service data in which 'language spoken' is recorded, we also included Tamil-speaking women who were born in Sri Lanka. The Tamils have a long history of persecution, with significant numbers coming to Australia on humanitarian grounds.

There was little change in the proportion of women of refugee background born in specific countries over the 3 years of data collection, as can be seen in S1 Table. Data from both hospital networks for the latter half of 2015–2016 indicated that 11%–13% of refugee women had arrived in Australia within the 2 years prior to the index birth.

Of the women of refugee background at hospital network X, over 40% were recorded as requiring an interpreter. The proportion of women of refugee background at hospital network Y requiring an interpreter was less than 25%, although data on this item were not recorded for a quarter of the women of refugee background attending this site (Table 2).

Table 3 reports the demographic and obstetric characteristics of Australian-born women and women of refugee background giving birth at the two hospital networks. Although the mean age at the birth of their child was similar for women of refugee background and Australian-born women within each hospital network, at hospital network X, a greater proportion of Australian-born women were young (14–19 years of age) compared with women of refugee background. At both hospital networks, women of refugee background were significantly more likely to be having their second or subsequent child (75%), compared with around 56% of Australian-born women. Around 15%–20% of women of refugee background had four or more children, five times the proportion Australian-born women. At hospital network X, a higher proportion of women of refugee background had gestational diabetes, and at both hospital networks, women of refugee background had lower reported preeclampsia/hypertension than Australian-born women.

### Attendance at antenatal visits

At the time of the first baseline measure in 2014 at hospital network X, around 60% of Australian-born women and women of refugee background attended seven or more antenatal visits, rising to around 75% at the second baseline measure and to 80% in 2015–2016 (see Fig 1 and S2 Table).

Multivariable logistic regression analyses accounting for clustering by country of birth within the hospital network, and for parity and gestational diabetes, estimated a linear trend whereby the odds of attending seven or more visits increased by around 20% over each

**Table 2. Migration characteristics of women giving birth at participating hospital networks, 2014–2016.**

| Migration characteristics | Hospital network X | | Hospital network Y | |
|---|---|---|---|---|
| | *n* | % | *n* | % |
| **All women** | | | | |
| **Maternal country of birth** | | | | |
| Australian-born | 10,277 | 39.2 | 6,298 | 39.5 |
| Migrant—English-speaking background | 1,341 | 5.1 | 851 | 5.3 |
| Migrant—non-English-speaking background (nonrefugee) | 11,797 | 45.0 | 6,651 | 41.7 |
| Migrant—refugee background | 2,740 | 10.5 | 1,414 | 8.9 |
| Missing | 55 | 0.2 | 718 | 4.5 |
| **Total** | **26,210** | **100.0** | **15,932** | **100.0** |
| **Women of refugee background** | *n* = 2,740 | | *n* = 1,414 | |
| **Country of birth** | | | | |
| Afghanistan | 1,511 | 55.1 | 53 | 3.7 |
| Sudan | 424 | 15.5 | 507 | 35.9 |
| Burma (Myanmar) | 169 | 6.2 | 287 | 20.3 |
| Iran | 147 | 5.4 | 60 | 4.2 |
| Iraq | 100 | 3.6 | 84 | 5.9 |
| Sri Lanka (Tamil only) | 134 | 4.9 | 34 | 2.4 |
| Somalia | 0 | 0 | 81 | 5.7 |
| Ethiopia | 133 | 4.9 | - | - |
| Congo | - | - | 58 | 4.1 |
| Eritrea | 11 | 0.4 | 40 | 2.8 |
| Liberia | 35 | 1.3 | 40 | 2.8 |
| Other | 76* | 2.8 | 170** | 12.0 |
| **Language spoken** | | | | |
| English | 917 | 33.5 | 708 | 50.1 |
| Dari | 860 | 31.4 | 7 | 0.5 |
| Pashto | 121 | 4.4 | 1 | 0.1 |
| Hazaragi | 84 | 3.1 | 3 | 0.2 |
| Persian (excluding Dari) | 111 | 4.1 | 56 | 4.0 |
| Arabic | 187 | 6.8 | 102 | 7.2 |
| Burmese | 107 | 3.9 | 122 | 8.6 |
| Karen | 7 | 0.3 | 16 | 1.1 |
| Rohingha | 34 | 1.2 | - | - |
| Chin Hakka | 2 | 0.1 | 83 | 5.9 |
| Dinka | 46 | 1.7 | 149 | 10.5 |
| Nuer | 36 | 1.3 | - | - |
| Tamil—Sri Lanka | 134 | 4.9 | 34 | 2.4 |
| Other | 94 | 3.4 | 83 | 5.9 |
| Missing | - | - | 50 | 3.5 |
| **Interpreter required** | | | | |
| No | 1,554 | 56.7 | 737 | 52.1 |
| Yes | 1,186 | 43.3 | 338 | 23.9 |
| Missing data | - | - | 339 | 24.0 |

*Other at hospital network X = Bhutan, Burundi, Cameroon, Guinea, Libya, Malawi, Mozambique, Rwanda, Sierra Leone, Tanzania, Uganda, Zambia, Syria.

**Other at hospital network Y = Benin, Burundi, Cameroon, Côte d'Ivoire, Guinea, Kenya, Rwanda, Sierra Leone, Tanzania, Togo, Uganda, Libya, Egypt, Syria, Bhutan, East Timor.

**Table 3. Obstetric characteristics of Australian-born women and women of refugee background giving birth at the two hospital networks, 2014–2016.**

| Obstetric characteristics | Hospital network X | | | | Hospital network Y | | | | |
| --- | --- | --- | --- | --- | --- | --- | --- | --- | --- |
| | Australian-born women | | Women of refugee background | | *p*-Value | Australian-born women | | Women of refugee background | | *p*-Value* |
| | Mean | SD | Mean | SD | | Mean | SD | Mean | SD | |
| **Maternal age** | 29.29 | 5.69 | 29.19 | 5.76 | 0.388 | 29.85 | 5.57 | 29.88 | 5.66 | 0.878 |
| | *n* | % | *n* | % | | *n* | % | *n* | % | |
| **Maternal age** | | | | | | | | | | |
| 14–19 years | 400 | 3.9 | 49 | 1.8 | <**0.001** | 203 | 3.2 | 30 | 2.1 | 0.106 |
| 20–24 years | 1,745 | 17.0 | 603 | 22.0 | | 1,043 | 16.6 | 248 | 17.5 | |
| 25–29 years | 3,180 | 30.9 | 828 | 30.2 | | 1,894 | 30.1 | 453 | 32.0 | |
| 30–34 years | 3,097 | 30.1 | 761 | 27.8 | | 1,909 | 30.3 | 417 | 29.5 | |
| 35+ years | 1,855 | 18.1 | 499 | 18.2 | | 1,249 | 19.8 | 266 | 18.8 | |
| **Parity**** | | | | | | | | | | |
| 0 | 4,532 | 44.1 | 751 | 27.4 | <**0.001** | 2,688 | 42.7 | 305 | 21.6 | <**0.001** |
| 1 | 3,402 | 33.1 | 698 | 25.5 | | 2,132 | 33.9 | 375 | 26.5 | |
| 2 | 1,510 | 14.7 | 535 | 19.5 | | 942 | 15.0 | 273 | 19.3 | |
| 3 | 513 | 5.0 | 345 | 12.6 | | 347 | 5.5 | 165 | 11.7 | |
| 4+ | 320 | 3.1 | 411 | 15.0 | | 189 | 3.0 | 296 | 20.9 | |
| **Gravida**\*\*\* | | | | | | | | | | |
| 1 | 3,143 | 30.6 | 612 | 22.3 | <**0.001** | 1,998 | 31.7 | 248 | 17.5 | <**0.001** |
| 2 | 2,987 | 29.1 | 615 | 22.4 | | 1,839 | 29.2 | 343 | 24.3 | |
| 3 | 1,863 | 18.1 | 497 | 18.1 | | 1,139 | 18.1 | 256 | 18.1 | |
| 4+ | 2,284 | 22.2 | 1,016 | 37.1 | | 1,322 | 21.0 | 567 | 40.1 | |
| **Gestational diabetes** | | | | | | | | | | |
| No | 9,463 | 92.1 | 2,287 | 83.5 | <**0.001** | 5,393 | 85.6 | 1,192 | 84.3 | 0.201 |
| Yes | 814 | 7.9 | 453 | 16.5 | | 905 | 14.4 | 222 | 15.7 | |
| **Hypertension/preeclampsia** | | | | | | | | | | |
| No | 9,941 | 96.7 | 2,715 | 99.1 | <**0.001** | 6,061 | 96.2 | 1,398 | 98.9 | <**0.001** |
| Yes | 336 | 3.3 | 25 | 0.9 | | 237 | 3.8 | 16 | 1.1 | |

*\*p*-Value from *t* test (for continuous age) or chi-squared test (for categorical variables) comparing the characteristics of Australian-born women and women of refugee background within each hospital network.

\*\*Women with a parity of 0 had no prior births (live-born or stillborn) and were expecting their first baby. Women with a parity of 1 had a single baby previously, parity of 2 indicated two previous babies, and so forth.

\*\*\* Women with a gravida of 1 were having their first pregnancy. Women with a gravida of 2 had a previous pregnancy and a current pregnancy, a gravida of 3 indicates two previous pregnancies and one current pregnancy, and so forth.

6-month period (adjusted odds ratio [adjOR] 1.22 [95% confidence interval (CI) 1.09–1.36], $p < 0.001$), but we did not find any additional evident difference comparing the intervention to baseline period (adjOR 1.07 [95% CI 0.91–1.27], $p = 0.413$) (see Table 4). There was little difference between the proportion of Australian-born women and women of refugee background having seven or more visits over time and no difference in terms of women having their first baby or second or subsequent baby (see Fig 1). No differential effects of time, intervention, parity, or gestational diabetes according to whether women were of refugee background or Australian-born were evidenced by tests of interaction (Table 4).

At hospital network Y, the first report of the number of antenatal visits by client commenced during the latter half of 2014. A quarter of the data were missing in the initial reporting period (baseline 2) at hospital network Y, with complete data in the final 18 months of the intervention period (June 2015 to December 2016; see Fig 2 and S2 Table).

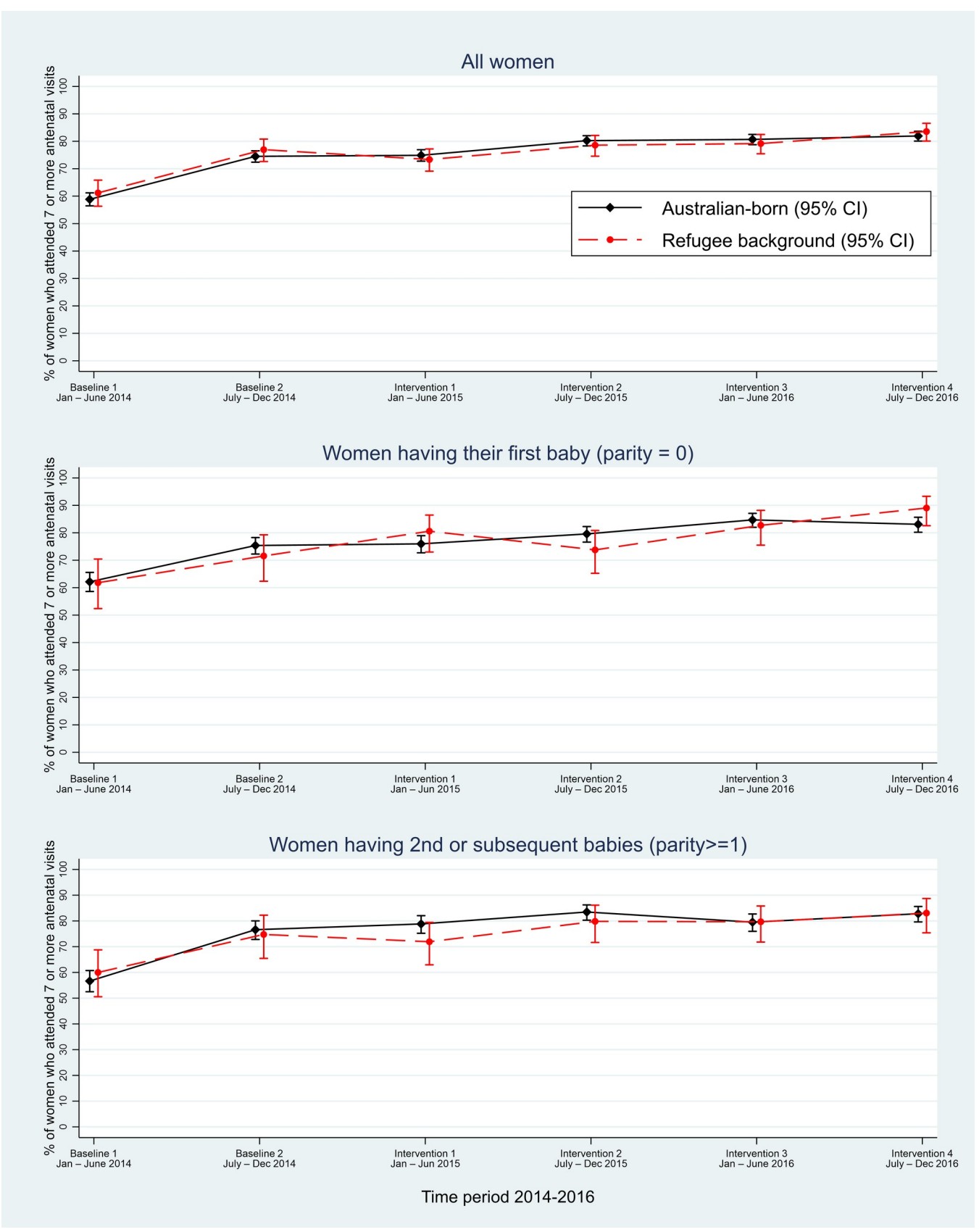

**Fig 1. Hospital network X: Proportion of women of refugee background and Australian-born women attending seven or more antenatal visits.** CI, confidence interval.

**Table 4. Regression analyses for hospital networks X and Y—Seven or more visits and gestation at first visit < 16 weeks.**

| Seven or more antenatal visits | | | | | | |
|---|---|---|---|---|---|---|
| Intervention, time, maternal characteristics | Hospital network X* | | | Hospital network Y** | | |
| | adjOR† | CI | *p*-Value | adjOR‡ | CI | *p*-Value |
| **Intervention** | | | | | | |
| Baseline 1 and 2 | 1.0 ref | | | 1.0 ref | | |
| Intervention 1–4 | 1.07 | 0.91–1.27 | 0.413 | 0.85 | 0.78–0.93 | <0.001 |
| **Time** | | | | | | |
| (per 6-month period) | 1.22 | 1.09–1.36 | <0.001 | 1.13 | 1.10–1.16 | <0.001 |
| **Country of birth** | | | | | | |
| Australian-born | 1.0 ref | | | 1.0 ref | | |
| Refugee background | 1.06 | 0.47–2.42 | 0.888 | 0.96 | 0.72–1.28 | 0.782 |
| **Parity** | | | | | | |
| Nulliparous | 1.0 ref | | | 1.0 ref | | |
| Multiparous | 0.85 | 0.81–0.89 | <0.001 | 0.97 | 0.93–1.02 | 0.253 |
| **Gestational diabetes** | | | | | | |
| No | 1.0 ref | | | 1.0 ref | | |
| Yes | 1.03 | 0.93–1.13 | 0.586 | 1.66 | 1.56–1.78 | <0.001 |
| **Gestation at first visit less than 16 weeks** | | | | | | |
| Intervention, time, maternal characteristics | Hospital network X*** | | | Hospital network Y**** | | |
| | adjOR† | CI | *p* | adjOR‡ | CI | *p* |
| **Intervention** | | | | | | |
| Baseline 1 and 2 | 1.0 ref | | | 1.0 ref | | |
| Intervention 1–4 | 1.06 | 0.78–1.44 | 0.693 | 1.27 | 1.11–1.44 | <0.001 |
| **Time** | | | | | | |
| (per 6-month period) | 0.97 | 0.93–1.01 | 0.093 | 0.88 | 0.86–0.89 | <0.001 |
| **Country of birth** | | | | | | |
| Australian-born | 1.0 ref | | | 1.0 ref | | |
| Refugee background | 0.89 | 0.32–2.48 | 0.825 | 0.60 | 0.44–0.82 | 0.001 |
| **Parity** | | | | | | |
| Nulliparous | 1.0 ref | | | 1.0 ref | | |
| Multiparous | 0.98 | 0.92–1.05 | 0.625 | 1.09 | 1.06–1.12 | <0.001 |

†Analyses used were logistic regression adjusted for clustering by hospital and maternal country of birth.

‡Analyses used were logistic regression adjusted for clustering by maternal country of birth.

*No differential effect of intervention (*p* = 0.144), time (*p* = 0.685), parity (0.832), or diabetes (*p* = 0.124) according to whether women were Australian-born or of a refugee background and attended seven or more antenatal visits at hospital network X.

**No differential effect of intervention (*p* = 0.712), time (*p* = 0.6), parity (0.146), or diabetes (*p* = 0.401) according to whether women were Australian-born or of a refugee background and attended seven or more antenatal visits at hospital network Y.

***No differential effect of intervention (*p* = 0.571), time (*p* = 0.779), or parity (0.819) according to whether women were Australian-born or of a refugee background and had a gestation of less than 16 weeks at first antenatal visit at hospital network X.

****No differential effect of intervention (*p* = 0.200), time (*p* = 0.327), or parity (0.831) according to whether women were Australian-born or of a refugee background and had a gestation of less than 16 weeks at first antenatal visit at hospital network Y.

Abbreviations: adjOR, adjusted odds ratio; CI, confidence interval; ref, reference

Analysis of available baseline data indicated that around 63% of both refugee background and Australian-born women had seven or more antenatal visits. There was some variation over the reporting period, with 72% of women of refugee background and 70% of Australian-born women attending seven or more visits at the last data collection point in 2016 and no difference in number of visits by parity (Fig 2). A steady linear trend reflecting an increase in odds of attending seven or more visits over time of around 10% per 6-month period was estimated (adjOR 1.13 [95% CI 1.10–1.16], $p < 0.001$), which was slightly offset by estimated overall decreased odds of attending seven or more visits during the intervention compared with the baseline period (adjOR 0.85 [95% CI 0.78–0.93], $p < 0.001$). No differential effects of time, intervention, parity, or diabetes, according to whether women were of refugee background or Australian-born, were evident (Table 4).

At baseline, two-thirds of Australian-born women and women of refugee background had their first hospital-based antenatal visit at <16 weeks' gestation at hospital network X, and 52% of women at hospital network Y. This proportion decreased over the period of observation. In the latter reporting period from July to December 2016, 64% of Australian-born women and 57% of women of refugee background at hospital network X had their first antenatal visit at less than 16 weeks' gestation (see S3 Table), although no statistically significant effects of time or intervention period were found (Table 4). At hospital network Y, 42% of Australian-born women and 34% of women of refugee background had their first antenatal visit at <16 weeks' gestation. The pattern of timing of visits was not associated with parity in that, proportionally, women having their first baby were just as likely to have their first visit at <16 weeks' gestation compared with multiparous women.

## Discussion

Bridging the Gap brought together public maternity hospitals, community-based services, researchers, and policy makers to codesign, implement, and evaluate a series of iterative innovations to improve maternal and child health outcomes for families of refugee background. To our knowledge, it is the first time codesigned organisational and systems change directed toward reducing health inequalities has been attempted in the universal platform of publicly funded maternity care.

One goal of the design and implementation of multiple initiatives within the whole-of-system programme was to improve refugee women's access to antenatal care. We sought to measure this in different ways designed to gauge the extent to which organisational and systems change resulted in improved accessibility and culturally appropriate service provision [24]. This paper reports on one element of our evaluation, which drew on routinely collected data to monitor changes over time in the number and timing of antenatal visits. The analyses showed that there were parallel trends of improvement in the proportion of women attending the recommended number of antenatal visits among both women of refugee background and Australian-born women. Counter to this, there was a steady decrease in the proportion of women having their first hospital-based antenatal visit in the first trimester of pregnancy. Despite the emphasis of Bridging the Gap initiatives on improving timely access to hospital-based antenatal care, this trend toward a decrease in the proportion of women having their first visit in the first trimester was apparent for women of refugee background and Australian-born women. We anticipated that this would result in women having fewer visits overall. The findings tell a different story. It appears that routine practice during the second and third trimesters of pregnancy may be changing.

Much has been written about implementation of interventions in complex systems of healthcare [16,25] including constantly changing context [26]. Anecdotally, hospital staff

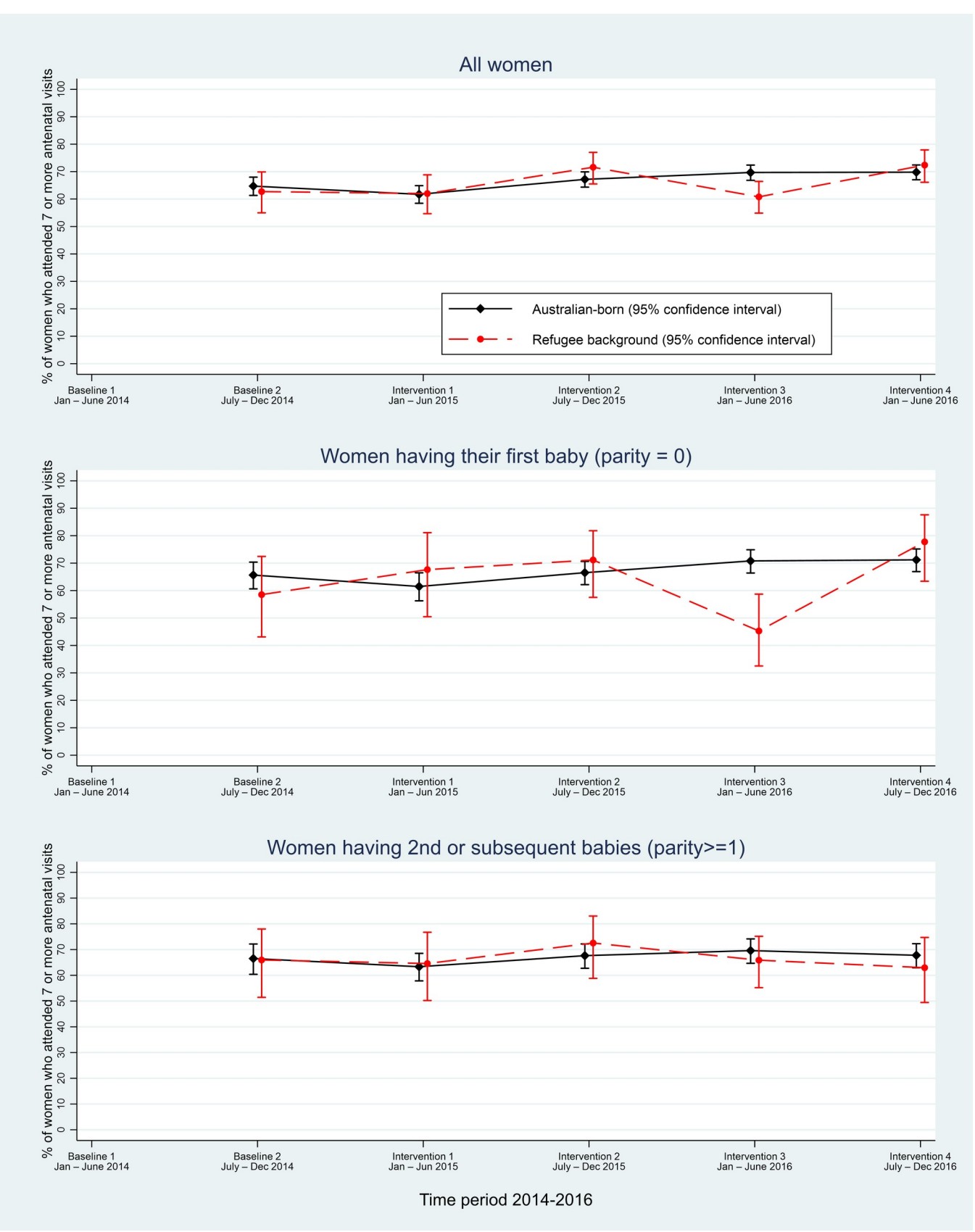

**Fig 2. Hospital network Y: Proportion of women of refugee background and Australian-born women attending seven or more antenatal visits.**

reported pressure to extend the gestation at the first hospital visit to the second trimester of pregnancy to manage increasing demand for care coupled with workforce shortages and no increase in resources or infrastructure over the time frame of the study. It is also likely that new guidelines for the management of gestational diabetes [3] and the release of a state government review of avoidable perinatal deaths at a Victorian health service [27] may have been factors in the apparent trend toward a greater proportion of women receiving seven or more visits. It appears that this may have been occurring at the expense of hospital-based care in the first trimester.

## Strengths and limitations

Strengths of this study include the interrupted time series design drawing on routinely collected data well suited to measuring change over time. The a priori decision to include Australian-born women as a comparison group enabled examination of trends for women known to have fewer barriers to accessing pregnancy care, with the additional benefit of assessing contextual factors likely to impact on all/both groups of women. A recent commentary highlights the potential advantages of a control group in which both the population of interest and the comparison/control group (with different characteristics to the controls) have been exposed to the same contextual events [27,28].

A further strength is the careful identification of women of refugee background using maternal country of birth combined with language spoken. The study illustrates the benefit of using routinely collected data in identifying women of refugee background.

We are mindful that the interrupted time series method in this study did not conform to all aspects of the standard design, including a smaller number of data points than the minimum of eight recommended by Penfold and Zhang [23]. Replication of the analysis using 3-month intervals resulted in minimal change to the findings but limited the number of refugee-background women within the comparisons at each time point. Given the binary outcome and substantial clustering effects of women's country of birth within each hospital network, which required robust estimation of standard errors, we were not able to account for the potential for autoregression whereby observations taken over time are correlated, as was also recommended by Penfold and Zhang [23].

We also recognise the inherent limitations of using the number of antenatal visits as the primary outcome measure for assessing the impact of the Bridging the Gap initiatives. This paper is just one arm of the Bridging the Gap evaluation. Other elements include tailored evaluation of individual quality improvement initiatives and demonstration projects. Qualitative insights into system change provide rich information for understanding the process and achievements of Bridging the Gap and will be reported in future papers.

## Implications for policy, practice, and research

Of the women giving birth at the four participating hospitals, 1 in 10 was of refugee background—double the original estimate. This has important policy implications. The underascertainment of refugee background due to poor systems for identification means that hospitals and governments are systematically underestimating the extent to which hospital services need to adjust and tailor care to the complex needs of specific populations.

Limitations of the interrupted time series design rather than randomisation [29] coupled with gaps in data availability mean that we are unable to determine the extent to which

Bridging the Gap initiatives influenced the overall improvement in access to antenatal care in the second and third trimesters of pregnancy. It is possible that greater disparities would have been evident had the Bridging the Gap programme not been implemented. However, we are also mindful that the codesigned initiatives were diverse, not all were focused on increasing access to visits, and the reach of initiatives was inevitably variable.

The apparent trend toward a decreasing proportion of women attending their first hospital visit in the first trimester of pregnancy is concerning. It is likely that women having their first hospital-based antenatal visit later in pregnancy are having some antenatal care provided by a GP, although this may not be the case for women unfamiliar with the Australian health system or who face language and other barriers in accessing primary care [30]. For women of refugee background, delayed access to hospital-based care increases the likelihood of missing out on critical elements of early pregnancy care (e.g., early screening tests) and/or other components of care (e.g., psychosocial assessment and support) that are vital to optimising maternal and child health outcomes.

Finally, the extent of missing data on interpreter requirements suggests that many women with low English proficiency may not be offered an interpreter, placing them at risk of missing out on crucial information about their health and that of their baby. As a quality and safety signal, missing data on the need for language services also means that the level of need for professional interpreter support is being underestimated.

## Conclusions

The study provides information to inform future health system reform measures. Importantly, we identified that the proportion of women of refugee background giving birth at the four study hospitals was far higher than anticipated. Accurate ascertainment of harder-to-reach populations and ongoing monitoring of the impact of quality improvement initiatives on populations vulnerable to poor health outcomes are essential to understand the impact of system reforms and efforts to reduce health inequalities. Our findings suggest that for both women of refugee background and Australian-born women, improvement in the total number of antenatal visits may have been at the expense of timely access to public hospital antenatal care in the first trimester.

## Supporting information

**S1 RECORD Checklist.**
(DOCX)

**S1 Fig. Data flowchart.**
(TIFF)

**S1 Table. Country of birth for women of refugee background by year for hospital networks X and Y.**
(DOCX)

**S2 Table. Mean antenatal visits by time period comparing Australian-born women and women of refugee background.**
(DOCX)

**S3 Table. Number antenatal visits and gestation at first visit by Australian-born women and women of refugee background over baseline (B) and intervention (I) time periods for hospital networks X and Y.**
(DOCX)

## Acknowledgments

We acknowledge members of the Bridging the Gap investigator team (Euan Wallace, John Furler, Rhonda Small, I-Hao Cheng) and steering group (Bernie Harrison, Pauline Petschel, Carol McIntyre, Colleen Turner, Natalija Nesvadba) for their contribution to the partnership. Bridging the Gap was funded by the National Health and Medical Research Council, and we acknowledge the in-kind contribution of the participating agencies and the support of the Victorian Government Infrastructure Fund. We acknowledge with thanks staff at the participating agencies who contributed to the achievements of Bridging the Gap and lessons learnt in 'doing things differently' in maternity and early child healthcare.

## Author Contributions

**Conceptualization:** Jane Yelland, Elisha Riggs, Josef Szwarc, Sue Casey, Stephanie J. Brown.

**Data curation:** Jane Yelland, Ellie McDonald, Dannielle Vanpraag.

**Formal analysis:** Jane Yelland, Fiona Mensah, Ellie McDonald, Dannielle Vanpraag.

**Funding acquisition:** Jane Yelland, Elisha Riggs, Josef Szwarc, Stephanie J. Brown.

**Investigation:** Jane Yelland, Fiona Mensah, Elisha Riggs, Josef Szwarc, Sue Casey, Christine East, Mary Anne Biro, Glyn Teale, Sue Willey, Stephanie J. Brown.

**Methodology:** Jane Yelland, Fiona Mensah, Elisha Riggs, Ellie McDonald, Josef Szwarc, Wendy Dawson, Dannielle Vanpraag, Sue Casey, Christine East, Mary Anne Biro, Glyn Teale, Sue Willey, Stephanie J. Brown.

**Project administration:** Jane Yelland, Elisha Riggs, Wendy Dawson, Dannielle Vanpraag.

**Supervision:** Jane Yelland, Elisha Riggs.

**Writing – original draft:** Jane Yelland, Fiona Mensah, Elisha Riggs, Ellie McDonald.

**Writing – review & editing:** Jane Yelland, Fiona Mensah, Elisha Riggs, Ellie McDonald, Josef Szwarc, Wendy Dawson, Dannielle Vanpraag, Sue Casey, Christine East, Mary Anne Biro, Glyn Teale, Sue Willey, Stephanie J. Brown.

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
