## [Decision Letter · Decision Letter 0]

27 Jan 2020

Dear Dr. Yelland,

Thank you very much for submitting your manuscript "‘Bridging the Gap’: Systems reform in Victorian public hospitals to improve access to antenatal care for women of refugee background evaluated using an interrupted time series design" (PMEDICINE-D-19-03494) for consideration at PLOS Medicine. 

[LINK]

In light of these reviews, I am afraid that we will not be able to accept the manuscript for publication in the journal in its current form, but we would like to consider a revised version that addresses the reviewers' and editors' comments. Obviously we cannot make any decision about publication until we have seen the revised manuscript and your response, and we plan to seek re-review by one or more of the reviewers. 

We expect to receive your revised manuscript by Feb 10 2020 11:59PM. Please email us (plosmedicine@plos.org) if you have any questions or concerns.

We look forward to receiving your revised manuscript. 

Sincerely,

Adya Misra, PhD

Senior Editor 

PLOS Medicine

plosmedicine.org

Title: Please revise to adhere to PLOS Medicine style, include a colon and a study descriptor in the second half the title. For example “Evaluation of systems reform in Victorian public hospitals to improve access to antenatal care for women of refugee background: an interrupted time series design” 

Abstract: please organise the abstract into “Introduction”, “Methods and Findings” followed by “Conclusions”. The last sentence of the methods and findings section should be a limitation of your methodology 

Abstract: When reporting quantitative results, please be specific instead of using “about” or “around”

Abstract: please include participant demographics 

Data availability statement- authors cannot be responsible for requests for data access and should be a third party such as an ethics committee 

Please remove page numbers from the RECORD checklist as these are likely to change. Instead please use paragraphs and sections

Did your study have a prospective protocol or analysis plan? Please state this (either way) early in the Methods section.

c) In either case, changes in the analysis—including those made in response to peer review comments—should be identified as such in the Methods section of the paper, with rationale.

Comments from the reviewers:

Reviewer #1: I confine my remarks to statistical aspects of this paper. The general approach is fine but I have one suggestion and a few concerns that need to be addressed before I can recommend publication.

The authors use logistic regression. This isn't exactly wrong, but, by dichotomizing the outcomes they lose power and lose some ability to answer interesting questions. I suggest using some sort of count regression (e.g. negative binomial) model for the first aim and Cox models for the second aim. 

In addition, the authors need to assess the assumptions of ITS analysis. In particular 1) Linear trend prior to intervention. Unless I am missing something, there are only 2 time points before the initervention, so this can't be assessed, but the assumption has to be mentioned. 2) That the characteristics of the population do not change over the course of the study. 

More specific:

Table 3 - I don't think age should be categorized; certainly not this way. Maybe use some sort of density plot, or maybe combine the youngest and oldest into a "high risk" category Also it isn't clear what the p value here is testing. Chi-square? Some sort of trend measure? 

Figures - making the y axis go from 0 to 100 does show the full possible range, but it obscures the trends. Consider using 50 to 100, 

Peter Flom.

Reviewer #2: Reviewer Report

Manuscript Title: 'Bridging the Gap': Systems reform in Victorian public hospitals to improve access to antenatal care for women of refugee background evaluated using an interrupted time series design

The manuscript is an insightful study based on two sets of expectant women, one - who sought refuge in Australia compared to two - native Australian women employing an interrupted time series analysis. The study is detailed and systematic with regard to the reported findings and results. The author(s) have employed a multivariate logistic regression model to determine the objectives of the study as well. They have mindfully reported the weaknesses of the study, especially with the limited time points employed in the time series. The findings procured with the multivariate logistic regression model spotlights answers to the research questions that the study attempts to investigate. 

The following points incorporated in the forthcoming draft would further enrich the contents of the manuscript enabling the reader with a better perspective of the study:

1. The manuscript has concise information about the 'Bridging the Gap' (BG) initiative which is the principal intervention of the study. 

a. Creating a new Background Information section that includes the aims, objectives, purpose, target metrics and milestones, vision and goals of the BG initiative would help the reader draw sufficient information about the BG initiative. This step is important because the BG initiative needs to be described adequately as the context of the paper revolves around it. 

2. The draft makes a mention of the hospitals that the two sets of expectant mothers visited for the study; however, it does not mention the inclusion and exclusion criteria of the selected hospitals.

a. Adding the inclusion and exclusion criteria of the hospitals in the Methods section is essential to further inform the reader about the demographics of the sample population of expectant mothers who visited the hospitals leveraged for the study.

Overall, I would recommend acceptance of the manuscript with incorporations of the above two points. 

Sincerely, 

Dr. Shenoy

Dr. Amrita G. Shenoy, PhD, MBA, MHA, MSc

Assistant Professor and Graduate Program Director

Health Systems Management Program

School of Health and Human Services

College of Public Affairs

University of Baltimore

1420 N. Charles Street, Baltimore, Maryland, 21201 USA

Reviewer #3: This is a fascinating study where the authors analyze the population of women receiving antenatal care at two hospital networks and try to assess differences based on refugee/immigration status. It would be interesting to try to tie this information to outcomes such as stillbirths, premature deliveries, obstetrical complications, maternal deaths. Could the authors do this?

could the authors comment on the link between antenatal care and outcomes in general.

The authors note that there is a higher incidence of poor birth outcomes amongst refugees yet by their data, there is a similar rate of antenatal visits of refugee women and Australian born women. Could the authors comment about this?

[LINK]

---

## [Decision Letter · Decision Letter 1]

19 Feb 2020

Dear Dr. Yelland,

Thank you very much for re-submitting your manuscript "Evaluation of systems reform in Victorian public hospitals to improve access to antenatal care for women of refugee background: an interrupted time series design" (PMEDICINE-D-19-03494R1) for review by PLOS Medicine.

I have discussed the paper with my colleagues and the academic editor and it was also seen again by xxx reviewers. I am pleased to say that provided the remaining editorial and production issues are dealt with we are planning to accept the paper for publication in the journal.

[LINK]

We look forward to receiving the revised manuscript by Feb 26 2020 11:59PM. 

Sincerely,

Adya Misra, PhD

Senior Editor 

PLOS Medicine

plosmedicine.org

Requests from Editors:

Title – can Australia be worked in please? 

Abstract - I think some more summary demographic information in the abstract may be useful including mean age and add months to the dates. Please add not sufficiently powered to limitations? (as in line 207)

Author summary -please include bullet points 

Author summary- lines 41 and 42 perhaps not needed here

Please provide hospital names participating in Victoria so the study findings are more relevant. Please consider including a map of Victoria indicating the spread of sites participating, city names in Victoria if hospital names could be potentially identifying.

Lines 44,45 we suggest you revise to “we co-designed and implemented multiple quality improvement and demonstration initiatives in universal health services, including four maternity hospitals” 

Lines 48-49- perhaps not relevant here? Suggest removing and incorporating in a previous point

Line 58-60 perhaps not relevant here and we suggest removing

Is this a significant limitation that could cause overestimation? Please discuss in the conclusions as required : Country of birth was considered the best available proxy measure for identifying women of refugee background in routinely collected hospital data systems

References- when citing multiple sources, please use single brackets for example Line 356 [27,28]

Data – "some restrictions", please indicate what these restrictions (ethical, consent related etc) are and if data might be available on request. Please note that authors cannot be data contacts. 

Comments from Reviewers:

Reviewer #1: The authors have responded to most of my issues satsifactorily. One remain issue. REgarding table 3, I wrote, asking them not to categorize age. They responded:

<<<

The categorisation helps with accessibility of the findings and allows for non-linearity in either risk or access according to age. As noted the youngest and oldest women would often experience the highest risks in

epidemiological studies but this may be due to very different social, biological and behavioural profiles so it would be a preference to keep these groups distinct

>>>

This is incorrect. Categorization often *obscures* nonlinearities, rather than illuminating them. I show some of this in my blog post https://medium.com/@peterflom/what-happens-when-we-categorize-an-independent-variable-in-regression-77d4c5862b6c

Peter Flom

[LINK]

---

## [Editor Report · Decision Letter 2]

13 Mar 2020

Dear Dr. Yelland,

Thank you very much for submitting your manuscript "Evaluation of systems reform in Victoria, Australia, public hospitals to improve access to antenatal care for women of refugee background: an interrupted time series design" (PMEDICINE-D-19-03494R2) for consideration at PLOS Medicine. 

[LINK]

Your response to comments from the statistical reviewer do not meet our requirements for publication and therefore we ask that you present age as a continuous variable, not a categorical variable. We will then seek additional review of the manuscript after you have undertaken the necessary revisions. 

In light of these reviews, I am afraid that we will not be able to accept the manuscript for publication in the journal in its current form, but we would like to consider a revised version that addresses the reviewers' and editors' comments. Obviously we cannot make any decision about publication until we have seen the revised manuscript and your response, and we plan to seek re-review by one or more of the reviewers. 

We expect to receive your revised manuscript by Apr 03 2020 11:59PM. Please email us (plosmedicine@plos.org) if you have any questions or concerns.

We look forward to receiving your revised manuscript. 

Sincerely,

Adya Misra, PhD

Senior Editor 

PLOS Medicine

plosmedicine.org

Comments from the reviewers:

[LINK]

---

## [Decision Letter · Decision Letter 3]

19 May 2020

Dear Dr. Yelland,

Thank you very much for re-submitting your manuscript "Evaluation of systems reform in Victoria, Australia, public hospitals to improve access to antenatal care for women of refugee background: an interrupted time series design" (PMEDICINE-D-19-03494R3) for review by PLOS Medicine.

I have discussed the paper with my colleagues and the academic editor and it was also seen again by xxx reviewers. I am pleased to say that provided the remaining editorial and production issues are dealt with we are planning to accept the paper for publication in the journal.

[LINK]

We look forward to receiving the revised manuscript by May 26 2020 11:59PM. 

Sincerely,

Adya Misra, PhD

Senior Editor 

PLOS Medicine

plosmedicine.org

Requests from Editors:

Please format the bibliography in Vancouver style

Please use paragraph and section numbers in the RECORD checklist instead of using page numbers as these are likely to change during publication. Could you also rename the file to read "RECORD" instead of "STROBE"

Comments from Reviewers:

Reviewer #1: The authors have responded to my concerns and I now recommend publication

Peter Flom

[LINK]

---

## [Editor Report · Decision Letter 4]

16 Jun 2020

Dear Dr. Yelland, 

On behalf of my colleagues and the academic editor, Dr. Paul Spiegel, I am delighted to inform you that your manuscript entitled "Evaluation of systems reform in public hospitals, Victoria, Australia, to improve access to antenatal care for women of refugee background: an interrupted time series design" (PMEDICINE-D-19-03494R4) has been accepted for publication in PLOS Medicine. 

PRODUCTION PROCESS

PRESS

PROFILE INFORMATION

Thank you again for submitting the manuscript to PLOS Medicine. We look forward to publishing it. 

Best wishes, 

Adya Misra, PhD

Senior Editor 

PLOS Medicine

plosmedicine.org